# Handwriting in Autism Spectrum Disorder: A Literature Review

**Henriette C. Handle [1,*], Marcus Feldin [1] and Artur Pilacinski [1,2,3]**

1    Faculty of Psychology, University of Warsaw, 00-183 Warsaw, Poland
2    Medical Faculty, Ruhr University Bochum, 44892 Bochum, Germany
3    Faculty of Psychology and Educational Sciences, University of Coimbra, 3001-802 Coimbra, Portugal
*    Correspondence: h.handle@student.uw.edu.pl

**Abstract:** Handwriting is linked to a variety of systems in the human brain and has been likewise demonstrated to be affected by a variety of neurological and developmental disorders. In this paper we provide a narrative review of recent findings regarding the quantitative evaluation of handwriting product in people with autism spectrum disorder. We summarize the experimental approaches and variables measured by most representative studies, such as handwriting speed and quality. We highlight the key issues such as small sample sizes resulting in underpowered designs. Lastly, we draw conclusions and delineate potential research directions, such as the use of machine learning to evaluate multivariate components of handwriting.

**Keywords:** autism; handwriting; motion tracking; developmental neuroscience





## 1. Introduction

Autism spectrum disorder (ASD) is a neurodevelopmental disorder which encompasses a broad range of complex developmental and neurobiological disabilities [1].The *Diagnostic and Statistical Manual of Mental Disorders* (DSM-IV) specifies impaired communication and social interaction, restricted interests, repetitive patterns in behavior, deficits in developing and maintaining relationships as well as impaired sensory information processing. In terms of severity the disorder spans from low to high functioning, both in terms of intelligence level and symptoms [2]. Handwriting is known to be challenging for many individuals on the spectrum due to difficulties with fine motor skills, and according to Cartmill and colleagues this difficulty is the main reason why as many as 86% of children diagnosed with ASD are referred to therapy services to improve their handwriting and fine motor skills [3].

Handwriting is typically assessed both through its handwritten product and handwriting process itself, such as speed of production. The handwritten product can be measured with regard to "readability" and "legibility" [4]. The development of computerized software and digital tablets has enabled quantitative measuring of these processes, whereas in the past, the handwritten product would be compared to global evaluation scales and normative samples. Today, researchers use more analytically based computerized evaluations which consider specific aspects of the "readability" and "legibility" criteria [5]

The criteria for the handwritten product are typically: form of the letter, sizing, spacing, line-straightness and consistency and are increasingly becoming standard as a consensus is being reached among the researchers who develop these analytical writing scales [5]. While examining the handwritten product of interest, the process in itself is also an area of interest as it may offer further insight about the writer's handwriting characteristics and potential challenges [4]. The process encompasses, for example, velocity, acceleration, direction, and changes in force [4].

With this literature review, we sought to examine the most current research available on the topic of handwriting quality and handwriting speed within the population of individuals with the Autism Spectrum Disorder.

## 2. Literature Review Summary

We reviewed six studies focusing on the topic of handwriting (e.g., handwriting quality, handwriting speed) in those with autism spectrum disorder. We selected the studies by browsing the Pubmed database with keywords "handwriting" and "autism". The studies were conducted in a similar manner, with no randomization and with comparisons between test and control groups. The lack of randomization occurred due to the inclusion criteria where the test group was required to be diagnosed with autism spectrum disorder. Autism spectrum disorder will be referred to as ASD throughout this text. The studies are listed and summarized in Table 1 below.

**Table 1.** A summary of the reviewed studies examining handwriting in ASD in terms of sample sizes, primary outcomes and study design.

| Study | No. of Participants | Outcome Domains | Study Design |
|---|---|---|---|
| Fuentes, Mostofsky and Bastian, 2009 | 14 ASD<br>14 Control | Handwriting quality | Non-randomized, between-groups with control group |
| Johnson, Papadopoulos, Fielding, Phillips and Rinehart, 2011 | 11 HF-ASD<br>11 Asperger<br>11 Control | Handwriting quality<br>Handwriting speed | Non-randomized, between-groups with control group |
| Hellinckx, Roeyers and Van Waelvelde, 2013 | 70 ASD<br>61 Control | Handwriting quality<br>Handwriting speed | Non-randomized, between-groups with control group |
| Rosenblum, Simhon and Gal, 2016 | 30 HF-ASD<br>30 Control | Handwriting quality | Non-randomized, between-groups with control group |
| Li-Tsang, Li, Ho, Lau and Leung, 2018 | 15 ASD<br>174 Control | Handwriting quality<br>Handwriting speed | Non-randomized, between-groups with control group |
| Godde, Tsao, Gepner and Tardif, 2018 | 21 ASD<br>42 Control | Handwriting quality<br>Handwriting speed<br>Predictors | Non-randomized, between-groups with control group |

## 3. Children with Autism Show Specific Handwriting Impairments

In a study by Fuentes, Mostofsky and Bastian [6], the authors sought to explore specific aspects of handwriting in which children with autism show difficulties, an area of research which until that date remained largely unexamined. A total of 28 participants, 14 with ASD and 14 typically developing controls, were first administered the WISC-IV (Wechsler Intelligence Scale for Children-IV) where full-scale IQs greater than 80 was observed in all but two subjects who showed marked discrepancies. The Perceptual Reasoning Index (the PRI) was used as the primary intelligence measure since the study involved nonverbal, perceptually based, motor tasks. Subjects were then administered the Minnesota Handwriting Assessment where they were asked to copy several words onto a provided solid line, making their letters the same size as the sample and using their best handwriting (Fuentes, Mostofsky and Bastian, 2009). The sample was then scored on each letter individually based on five categories, namely: alignment, legibility, size, form and spacing. Motor skills were assessed using the Revised Physical and Neurological Examination for Subtle (Motor) Signs (PANESS) [6]. This measure comprises several categories such as balance and timed movements, heel, and toe walking, hopping on one foot and finger apposition. The subjects were also asked to undergo the Block Design test, which is a subtest of the WISC-IV that assesses visuospatial abilities. In this test, the

subjects were asked to reconstruct several advanced designs by way of assembling a set of blocks which are components of a larger pattern.

The results showed that, although the children diagnosed with ASD performed worse in the quality of forming letters, neither of the groups demonstrated differences in relation to alignment, size, and spacing. No significant differences were found between groups in terms of age, Block Design, or PRI score. However, consistent with previous studies, the control group performed better overall on the PANESS inventory, specifically in the gait/stances and timed movements subcategories. Stepwise multiple regressions analysis revealed that the PANESS timed movements scores were the strongest predictors when it comes to handwriting performance in the ASD group [6].

## 4. Handwriting in Children with Autism and Asperger's Disorder

A study by Johnson and colleagues aimed to compare and investigate the handwriting profile of children diagnosed with either high-functioning autism spectrum disorder or Asperger's [7]. The study included three groups with 11 subjects in each group: high-functioning ASD, Asperger's, and a control group. The participants were first asked to fill out the Wechsler Intelligence Scale for children-IV and were subsequently matched according to age and Perceptual Reasoning Index (the PRI) score. The participants were then asked to perform three different writing tasks. Under three different conditions, they would be asked to write a cursive letter "l" on a digital tablet, in different sizes, with five trials per condition. With use of special software, the authors of the study were able to extract kinematic and temporal features, e.g., height, length, width, duration, and pen pressure. The participants then completed a speed subtest from the Handwriting Performance Test to assess change in handwriting over time, as well as handwriting speed. Here they were tasked with writing the words "cat and dog" as many times as they could on a specific line for a duration of two minutes. The height, spacing and width of each of these words, as well as the whole phrase, was then measured and scored [7].

Perhaps the most important finding from this study was that once the participants of the clinical groups were faced with the lack of visual cues, their sizing of the letters increased significantly, which suggests that handwriting size is motored by contextual and visual guides [7]. Decreased space and increased variability in spacing were found between words on the "cat and dog" task in the clinical groups, which could contribute to their under-average handwriting legibility. Overall, few differences were found between the two clinical groups, with the control group performing better in all tasks. Although these two conditions are clinically distinct from one another, there is significant overlap between the neurobiological and clinical symptomatology. This study demonstrated support to the revisions of DSM-V, in which ASD and AD have been merged into the "autism spectrum disorder" category. The study also confirmed the finding that children with these neurobiological conditions perform better when provided with visual cues as guidelines, which serves as important knowledge for those working within educational and/or therapeutic settings as they develop strategies for improving handwriting in these populations [7].

## 5. Predictors of Handwriting in Children with Autism Spectrum Disorder

The study conducted by Hellinckx, Roeyers and Van Waelvelde aimed to investigate several factors of handwriting quality and speed in children with autism spectrum disorder [8]. The 131 participants completed IQ tests, with FSIQ and WISC-III; 70 of the children were previously diagnosed with ASD and 61 children were typically developing. To measure handwriting quality and speed, the Dutch Systematic Screening of Handwriting tool was used. This tool detects graphomotor disorders in children. Participants were instructed to copy a text as fast and as neatly as they could onto an unruled paper for five minutes or until five sentences were completed. The sentences are used to measure quality through fluency of letter formation, fluency in connecting letters, letter height, regularity of letter height, spaces between words, and spatial alignment of sentences. Handwriting speed was measured by counting the number of letters written in five minutes. Scores below the 5th

percentile indicate graphomotor disorder. The children were additionally tested with the M-ABC-2, which measures motor functioning and identifies movement difficulties, as well as the VMI which measures visual-motor integration skills. They were also instructed to complete the One Minute Reading Test which measures the amount of correctly read words for one minute [8].

The results of the study indicate that children with ASD perform poorer on all measures. Children with ASD had lower handwriting quality compared to the control group, as they had difficulties with connecting letters, wrote less fluently, and had irregularities in height and spatial alignment. Additional findings include that handwriting quality is better in higher ages, as well as boys having poorer quality of handwriting than girls. No difference was found between left- and right-handed writing. In the population with ASD, handwriting quality could be predicted with age, gender and the VMI scores. Additionally, handwriting speed was correlated with age. The study successfully identified three predictors of handwriting quality and speed: age, gender, and VMI results. The study also shows that participants with ASD have poorer handwriting skills, as the quality and speed of their handwriting is poorer than the neurotypical participants. However, better coordination between visual input and finger movement resulted in higher quality output in those with ASD, indicating that improving these skills may result in better performance on handwriting quality and handwriting speed tests [8].

### 6. Unique Handwriting Performance Characteristics of Children with High-Functioning Autism Spectrum Disorder

The aim of the study by Rosenblum, Simhon and Gal was to compare the product characteristics and handwriting process of children who have been diagnosed with high-functioning autism spectrum disorder (HF-ASD) with a control group consisting of typically developing children, to find the best way of differentiation between the two groups [1]). The participants were 60 children between 9 and 12 years old; 30 who were diagnosed with HF-ASD, and 30 who were age- and gender-matched controls. They were asked to perform three different writing tasks on a digital tablet, which used the computerized handwriting evaluation system (ComPET). The tasks consisted of the participants writing their name and surnames, copying a paragraph, and writing a story based on a picture they were provided with—so-called free-style writing [1]. Upon completion, the participants' paragraph copying result was then assessed with the use of the Hebrew Handwriting Evaluation (HHE), which is a standardized test that comes with a rating of global legibility (i.e., the overall clarity of handwriting), the number of letters erased and/or overwritten, the number of unrecognizable letters and spatial arrangement [9]. The computerized handwriting evaluation system (ComPET) measured pen tilt, temporal measures in seconds, on-paper stroke duration, in-air stroke duration, stroke width and height, as well as pen pressure.

Significant differences were found across all tasks in relation to stroke time on-paper; however, mean stroke in-air was significantly different for the name task and the free-style writing tasks, but not for the paragraph copying task. This suggests that when a child diagnosed with HF-ASD is provided with a visual guideline such as an image or a paragraph, they invest the same amount of time planning the next pen stroke as the typically developed children do. When that image or paragraph is removed, the children diagnosed with HF-ASD need more time to produce, reconstruct and plan their handwriting. This could be explained by challenges with visual perception in individuals with HF-ASD [1]. Overall, the results showed significantly higher scores for the control group. Perhaps more importantly though, the results showed that both pen stroke duration in-air and on-paper helped predict the speed of handwriting, which could suggest that a child diagnosed with HF-ASD invests a lot of energy in the mechanical process related to producing handwriting [1].

### 7. The Relationship between Sensorimotor and Handwriting Performance in Chinese Adolescents with Autism Spectrum Disorder

The study conducted by Li-Tsang, Li, Ho, Lau and Leung aims to explore the connection between sensorimotor control and handwriting problems [10]. The control group consisted of 174 typically developing adolescents, and the test group consisted of 15 participants with diagnosed ASD. To measure handwriting, the participants completed the Computerized Handwriting Speed Test System (CHSTS-2). In this measure, the participants are asked to copy 130 Chinese characters and 120 English words on A4 paper connected to a tablet with an electronic pen. The test provides information on ground time (pen on paper), airtime (pen off paper), handwriting speed (character per minute), standard deviation of writing time per character, pen pressure, SD of pen pressure, and readability (correctness of words and number of words recognized by the system). The participants were also tested on motor skills, visual perceptual skills, visual-motor integration, and eye movements. These were tested through the measures the Bruininks-Oseretsky Test of Motor Proficiency (BOT), the Motor-Free Visual Perception Test (MVPT), the Beery-Buktenica Developmental Test of Visual-Motor Integration (VMI), and the Developmental Eye Movement test (DEM), respectively [10].

The results of the study show that participants with ASD had more ground and airtime, wrote more slowly, and had larger variations in writing speed than the neurotypical participants. They also showed less stability in their handwriting as can be seen through larger writing speed and pen pressure variations. Poorer manual dexterity was associated with ASD. In Chinese handwriting, poor manual dexterity increased ground time and reduced writing speed. In English handwriting, poor manual dexterity increased airtime and writing speed variation. Manual dexterity was found to be a significant mediator between ASD and Chinese handwriting, whereas ground time and writing speed were found to be a significant mediator between ASD and English handwriting. In terms of readability, participants with ASD show comparable results to the neurotypical participants, which is argued as resulting from specialized training in these aspects in childhood. This study indicates that handwriting results are consistent in both Chinese and English handwriting tasks, which suggests that the handwriting is influenced by ASD rather than the language skills of the participants. Manual dexterity is the main aspect found to affect handwriting results, which indicates that manual dexterity training might improve handwriting in adolescents with ASD [10].

### 8. Characteristics of Handwriting Quality and Speed in Adults with Autism Spectrum Disorders

The study conducted by Godde, Tsao, Gepner and Tardif aimed to explore features of handwriting and the role of perceptual-motor skills in adults with autism spectrum disorder [11]. The 63 participants were divided into three groups, the first being adults with ASD and the other two being control groups. The participants with ASD completed the Raven's Standard Matrices test with visuospatial reasoning, nonverbal intelligence tests and inductive-logical reasoning to identify the developmental age of the participants. The control groups consisted of non-ASD adults who matched the chronological age of the test group, and 21 non-ASD children who matched the developmental age of the test group. Handwriting quality and speed of the participants were measured with the Concise Evaluation Scale for Children's Handwriting (BHK). The participants are asked to copy a text for five minutes, or until the first five sentences are completed. These first five sentences are larger than the following sentences. The test results display scores on letter size, left margin widening, word alignment, word spacing, chaotic writing, irregularities in joining strokes, collision of letters, letter size, height of letters, letter distortion, ambiguous letter forms, correct letter forms, and unsteady writing. The participants were also assessed on their perceptual-motor skills through the Developmental Neuropsychological Assessment (NEPSY-1). Finger dexterity was measured through a finger-tapping task, fine motor coordination was measured through imitating hand or finger positions from a model,

graphomotor skills were measured through a figure copy task, and visuomotor integration was measured by drawing a line inside a path as quickly as possible without leaving the track [11]. The results of the study show that the participants with ASD performed poorer than both control groups. No differences between the control groups were found. Differences included poorer word alignment, ambiguous letter forms, marginal effects in left margin widening, insufficient word spacing, and inconsistent letter size. In terms of writing speed, the participants with ASD wrote more slowly than the chronological age group, but no difference was found between the participants with ASD and the developmental age group. In the participants with ASD, handwriting speed was significantly influenced by finger dexterity, graphomotor skills, and visuomotor integration. Developmental age was the best predictor for handwriting quality. The findings of the study highlight that handwriting difficulties in those with ASD persist throughout adulthood. Training skills such as finger dexterity, graphomotor skills, and visuomotor integration might improve handwriting quality and speed of those with ASD in adulthood [11].

## 9. Discussion

In each of the reviewed studies, researchers used similar designs, comparing handwriting capacities of participants diagnosed with ASD to control groups comprising neurotypical participants. This allows for comparisons between the results of the studies. In extension, it appears that most of these studies agree with each other. The studies show that the participants with ASD perform poorer on measures of handwriting quality and handwriting speed compared to typically developed participants. Four separate studies indicate that motor skills play a large part in handwriting in ASD participants, highlighting that a better score on factors such as manual dexterity, graphomotor skills, and general motor skills might aid ASD participants to perform better in terms of handwriting quality and speed. Moreover, it can be used as predictor of handwriting scores. Additionally, four of the studies found results supporting that visual input, visual perception, and visuomotor integration is important and might be a predictor for handwriting quality and speed in those with ASD. Improving motor skills, as well as visual perception and visuomotor integration, can be important factors in improving handwriting quality and speed in people diagnosed with ASD, and allowing people diagnosed with ASD to have visual input while performing tasks like the ones in the research can also improve the output.

Most of the studies focused on children diagnosed with ASD, however two of the studies focused on adolescents or adults diagnosed with ASD. These studies were added to the literature review to analyze whether the difficulties ASD participants have in the studies including children would persist throughout adulthood. Both studies that focused on adolescents or adults maintained that difficulties do persist. The first study focusing on adolescents found manual dexterity to be the main aspect that affects handwriting results. The second study found that not only manual dexterity, but also graphomotor skills and visuomotor integration are important aspects that influence handwriting in ASD participants in adulthood as well as in childhood. The persistence of these difficulties implies that improving training or support in these areas in childhood can improve handwriting outcomes in adults diagnosed with ASD.

It is worth noting that in chronological order, the studies in this field that focus on the topic of handwriting output in participants diagnosed with ASD improve with time. The earlier studies are less specific and focus on a broader horizon, while the later studies become narrower and more specific in learning which factors truly affect handwriting output. This implies that we have become more knowledgeable on the topic, and that we move closer and closer to learning about skills that can truly aid the population on the spectrum in improving handwriting and other variables that are affected by the same skills. An example of that may be such skills as playing instruments or drawing.

On a slightly critical note, one seemingly problematic factor which some of the authors have pointed out is the fact that very few of these studies use the same tools and inventories for measuring and scoring variables [4,5,12]. While different digital software recording, e.g.,

pen pressure or milliseconds of on-paper stroke duration, may not logically differentiate significantly between their produced results, they will, if programmed correctly, record the same result which one can use to compare between studies with minimal effort. Different tasks, and protocols, in the way they are provided to the participant, may present larger differences and biases in their measurements when attempting to compare such studies.

In the cited studies, the sample sizes are consistently small. Except for one study by Hellinckx, Roeyers and Van Waelvelde [8] with 70 ASD participants, none of the examined studies included more than 30 participants diagnosed with ASD, with an average of just 28.6 participants in our examined studies. One could argue that these numbers are too small to generalize to larger populations of individuals on the spectrum. The statistical significance of the results the researchers have found might be faulty or unreliable, due to the increased margin of error. At the same time, one also must ask how realistic and plausible larger sample sizes really are, taking into consideration the size of the population of interest.

Lastly, recent studies on ASD and the use of tablet devices identified distinct patterns of forces and gesture kinematics when children with ASD used tablets while playing serious games [13]. The authors concluded that children with autism showed generally higher contact forces and faster movements when playing the games. While these games did not directly employ handwriting, such combined analysis of kinematics and dynamics of movement might in the future yield a more complex and sensitive way of testing for ASD and other disorders affecting both cognition and motor control.

## 10. Conclusions

Autism spectrum disorder affects several skills pertaining to handwriting outcome, such as manual dexterity, graphomotor skills, general motor skills, visual input, visuomotor integration and visual perception. Additionally, visual cues or guidelines are shown to assist those with ASD in better handwriting outcomes. Some of the difficulties pertaining to these skills persist throughout adulthood in those with ASD. A possible recommendation for educators, parents, therapists, and rehabilitators would be to include specific training of these skills in ASD childhood, as well as using visual cues to assist in learning. These skills likely apply to other life skills or hobbies and improving these skills might increase the quality of life for those with ASD. Using visual cues with those with ASD is not an uncommon practice, but studies in this review support the continuation of this aspect.

Unfortunately, despite becoming more knowledgeable about the topic throughout the years, it seems there should be more of an agreement on which measures to use as somewhat of a standard when studying the topic. Using many different measures might lead to many different results that are not comparable to each other, therefore using the same measures would allow for better comparisons. The studies in this review have quite small sample sizes, likely due to accessibility of participants with ASD or not wanting to participate. Small sample sizes bring quite a few disadvantages. The results of these studies might not be applicable to the general population, and the statistical significance of the results may be faulty. A recommendation for further research would be to increase sample sizes, as well as improve and agree on a measurement procedure. Additionally, it could be helpful to the generalizability of these studies to attempt to randomize several test groups and compare these to control groups instead of only having one non-randomized test group. Further research should continue to be narrowed to specific aspects that might affect handwriting output in the autism spectrum disorder population.

**Author Contributions:** Conceptualization: H.C.H., M.F. and A.P.; Writing (original draft): H.C.H. and M.F.; Writing (review and editing): H.C.H., M.F. and A.P. All authors have read and agreed to the published version of the manuscript.

**Funding:** This research was funded by FCT project (PTDC/PSI-GER/30745/2017).

**Institutional Review Board Statement:** Not applicable.

**Informed Consent Statement:** Not applicable.

**Data Availability Statement:** Not applicable.

**Conflicts of Interest:** The authors declare no conflict of interest.

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
