# Peer review of "Handwriting in Autism Spectrum Disorder: A Literature Review"

_neurosci, doi:10.3390/neurosci3040040_

Round 1

Reviewer 1 Report

Overall, the literature review is compelling and informative. The text highlighted components of handwriting that may pose obstacles for individuals with ASD. Educators and therapists will be able to use this article to inform their practice by following the suggestions included in the discussion and conclusion components of the review. 

The work is well organized and easy to follow. Each article is described comprehensively and in words can be decoded by multiple audiences. A few spots could possibly benefit from expansion. Such areas are highlighted in the line by line review below. A few guiding questions have been attached to aid in extrapolation. Additionally, the work appears to be scientifically sound and avoids misleading the audience by contemplating possible weaknesses in the literature that is presented. Throughout the work there are several appropriate references to related pieces of research. One spot could possibly benefit from an explicit statement of authors’ names (see comment on line 298). 

English conventions and general readability of text was best surrounding discussion of the highlighted studies in parts 3-8. I found that in parts 1,2, and 9, extensive editing of English language and style was required. In the line-by-line comments, I suggest possible ways to rearrange sentence structures that may improve readability. Sentences throughout the text tended to have extraneous words and articles that inhibited the reader’s understanding. 

In reading this paper, I found no potential conflict of interest. I detected no plagiarism or inappropriate self-citations. All of the citations in the bibliography were found in the paper. It was a pleasure to read your literature review. I anxiously await your final product. 

Line 8: Delete the first sentence

Line 13: What is meant by “most experimental approaches and variables”?

Line 16: future directions is vague 

Line 20: referring to ASD as a neurological “disability” may come across as insensitive to individuals in the ASD community, especially as it is followed by the next line explaining some deficits that many people with ASD experience

Line 29: what sort of therapy services, occupational? 

Line 30-31: consider changing to -- Handwriting is typically assessed by examining physical processes, written products, and speed of performance.  

Line 31: Delete “the” 

Line 34: delete “the”, change to “hand written products” -- change wherever else found in the document

Line 35-36: Consider changing to: Today, researchers use more analytically based computerized evaluations to consider “readability” and “legibility” criteria. 

Line 37-40: I am not sure what is meant by this sentence

Line 41-45: consider changing to: Additionally, examining an individual’s handwriting process offers further insight into the writer’s potential challenges. The handwriting process encompasses, velocity, acceleration, direction, and changes in force. 

Line 37-40: I like that you explain what the criterion for written products and physical processes looks like, but I think this area could be expanded upon a little more. What specific things do examiners look for? What sort of challenges are unearthed through examination? For example, if a student is observed as having low velocity in their handwriting procedure, what sort of questions would that lead an examiner to ask? 

Line 48: consider changing to: … with Autism Spectrum Disorder 

Line 51-52: passive language, consider changing to, we reviewed six studies that examined the handwriting of individuals with ASD. 

Line 57: I am not sure what is meant by the sentence “some studies found predictors for ..”

Line 63-64: consider changing to-- Research exploring aspects of handwriting that are difficult for children with ASD remained largely unexamined until a study conducted in 2009 by Fuentes, Mostofsky & Bastian. 

Line 93: consider changing “the study” to “a study”, the first sounds like the study you cite is the only study that these authors have written (considering changing throughout the document) 

Line 132: “as fast and as neatly” 

Line 262-263: Consider changing to--  “In each study under review, researchers compare handwriting capabilities of individuals with ASD to control groups of neuro-typical participants. Consistency in study design allows for results of the studies to be compared.”

Line 264: delete “to that”

Line 265: Delete “the common theme here” consider combining with the previous sentence, “In extension, all studies seem to agree that participants with ASD…” 

Line 267: consider identifying which four studies 

Line 268-271: “... that motor skills pose an obstacle to the handwriting product and processes of individuals with ASD, highlighting that a increasing skill set in manual dexterity… might aid participants with ASD to perform better in terms of handwriting quality and speed” Delete “as well, it can be used as a predictor of handwriting scores” or introduce as a new thought. 

Line 274-275: a little redundant, consider deleting. 

Line 284: consider splitting up the sentence into two 

Line 289-296: I find this paragraph a little weak, I am not sure that it is necessary to say that as more studies are conducted, the more we begin to understand the issues

Linke 298: consider citing which authors have pointed out the critique

Line 306: consider creating adding a transitional word to the beginning of the paragraph to make it read smoother

Line 314-312: this paragraph comes off as a bit of an afterthought, I am not sure that it is totally necessary, but if you expand upon it, it could be an interesting aspect of the review

Author Response

Thank you for your compelling and exhaustive review and all the suggestions. We have appended the changes according to your comments on the basis of the list below.

Overall, the literature review is compelling and informative. The text highlighted components of handwriting that may pose obstacles for individuals with ASD. Educators and therapists will be able to use this article to inform their practice by following the suggestions included in the discussion and conclusion components of the review. 

The work is well organized and easy to follow. Each article is described comprehensively and in words can be decoded by multiple audiences.

Thank you for the appreciation

A few spots could possibly benefit from expansion. Such areas are highlighted in the line by line review below. A few guiding questions have been attached to aid in extrapolation. Additionally, the work appears to be scientifically sound and avoids misleading the audience by contemplating possible weaknesses in the literature that is presented. Throughout the work there are several appropriate references to related pieces of research. One spot could possibly benefit from an explicit statement of authors’ names (see comment on line 298). 

Fixed, thanks for spotting this.

English conventions and general readability of text was best surrounding discussion of the highlighted studies in parts 3-8. I found that in parts 1,2, and 9, extensive editing of English language and style was required. In the line-by-line comments, I suggest possible ways to rearrange sentence structures that may improve readability. Sentences throughout the text tended to have extraneous words and articles that inhibited the reader’s understanding. 

Thank you. We appended the changes as suggested.

In reading this paper, I found no potential conflict of interest. I detected no plagiarism or inappropriate self-citations. All of the citations in the bibliography were found in the paper. It was a pleasure to read your literature review. I anxiously await your final product. 

Line 8: Delete the first sentence

Line 13: What is meant by “most experimental approaches and variables”?

Line 16: future directions is vague 

We rephrased this to define that we specifically mean research directions

Line 20: referring to ASD as a neurological “disability” may come across as insensitive to individuals in the ASD community, especially as it is followed by the next line explaining some deficits that many people with ASD experience

We decided to leave the wording as is since "disability" is a neutral term used in the literature.

Line 29: what sort of therapy services, occupational? 

We clarifed this in line with the source

Line 30-31: consider changing to -- Handwriting is typically assessed by examining physical processes, written products, and speed of performance.  

We rephrased it differently as performance can be misleading

Line 31: Delete “the” 

Line 34: delete “the”, change to “hand written products” -- change wherever else found in the document

Line 35-36: Consider changing to: Today, researchers use more analytically based computerized evaluations to consider “readability” and “legibility” criteria. 

Done

Line 37-40: I am not sure what is meant by this sentence

We rephrased it now.

Line 41-45: consider changing to: Additionally, examining an individual’s handwriting process offers further insight into the writer’s potential challenges. The handwriting process encompasses, velocity, acceleration, direction, and changes in force. 

Line 37-40: I like that you explain what the criterion for written products and physical processes looks like, but I think this area could be expanded upon a little more. What specific things do examiners look for? What sort of challenges are unearthed through examination? For example, if a student is observed as having low velocity in their handwriting procedure, what sort of questions would that lead an examiner to ask? 

Thanks for pointing this out. We provide more detailed variables and conclusions further down the text so tried to keep the introduction succinct.

Line 48: consider changing to: … with Autism Spectrum Disorder 

Line 51-52: passive language, consider changing to, we reviewed six studies that examined the handwriting of individuals with ASD. 

Line 57: I am not sure what is meant by the sentence “some studies found predictors for ..”

Removed the sentence as not necessary

Line 63-64: consider changing to-- Research exploring aspects of handwriting that are difficult for children with ASD remained largely unexamined until a study conducted in 2009 by Fuentes, Mostofsky & Bastian. 

Line 93: consider changing “the study” to “a study”, the first sounds like the study you cite is the only study that these authors have written (considering changing throughout the document) 

Line 132: “as fast and as neatly” 

Line 262-263: Consider changing to--  “In each study under review, researchers compare handwriting capabilities of individuals with ASD to control groups of neuro-typical participants. Consistency in study design allows for results of the studies to be compared.”

Line 264: delete “to that”

Line 265: Delete “the common theme here” consider combining with the previous sentence, “In extension, all studies seem to agree that participants with ASD…” 

Line 267: consider identifying which four studies 

Line 268-271: “... that motor skills pose an obstacle to the handwriting product and processes of individuals with ASD, highlighting that a increasing skill set in manual dexterity… might aid participants with ASD to perform better in terms of handwriting quality and speed” Delete “as well, it can be used as a predictor of handwriting scores” or introduce as a new thought. 

Line 274-275: a little redundant, consider deleting. 

Line 284: consider splitting up the sentence into two 

Line 289-296: I find this paragraph a little weak, I am not sure that it is necessary to say that as more studies are conducted, the more we begin to understand the issues

We found it important to emphasize that we are dealing with an emerging field

Linke 298: consider citing which authors have pointed out the critique

Done.

Line 306: consider creating adding a transitional word to the beginning of the paragraph to make it read smoother

Line 314-312: this paragraph comes off as a bit of an afterthought, I am not sure that it is totally necessary, but if you expand upon it, it could be an interesting aspect of the review

Unfortunately, there is not much to write on this. Motor signatures of neurodevelopmental and neurodegenrative disorders are a very new field and there is a lack of coherent picture we could draw beyond the brief notion we drafted here.

Reviewer 2 Report

In the present manuscript, the authors have summarized the critical pieces of literature focused on the handwriting ability of the population diagnosed with Autism Spectrum Disorder (ASD) from childhood to adult samples in different studies. The summary and explanation of the case studies and their conclusions are defined in a good way. I also liked the discussion which was focused on the cumulative outcomes of the studies and crucial factors such as visual cues and others mentioned in the manuscripts that might be beneficial for training ASD children by educators, teachers, and parents.

I recommend the article should be accepted with minor language and spelling checks.

Author Response

Thank you so much for the appreciation. The newer version of the manuscript contains a number of language edits as thoroughly pointed out by Reviewer 1. We believe it now reads much better.